# Adaptive Batch Size Schedules for Distributed Training of Language Models with Data and Model Parallelism

Tim Tsz-Kit Lau*†‡    Weijian Li*§    Chenwei Xu§    Han Liu§    Mladen Kolar†¶

‡University of Pennsylvania  §Northwestern University  ¶University of Southern California

timlautk@upenn.edu; {weijianli2021,chenweixu2023}@u.northwestern.edu;
hanliu@northwestern.edu; mkolar@marshall.usc.edu

An appropriate choice of batch sizes in large-scale model training is crucial, yet it involves an intrinsic yet inevitable dilemma: large-batch training improves training efficiency in terms of memory utilization, while generalization performance often deteriorates due to small amounts of gradient noise. Despite this dilemma, the common practice of choosing batch sizes in language model training often prioritizes training efficiency—employing either constant large sizes with data parallelism or implementing batch size warmup schedules. However, such batch size schedule designs remain heuristic and often fail to adapt to training dynamics, presenting the challenge of designing adaptive batch size schedules. Given the abundance of available datasets and the data-hungry nature of language models, data parallelism has become an indispensable distributed training paradigm, enabling the use of larger batch sizes for gradient computation. However, vanilla data parallelism requires replicas of model parameters, gradients, and optimizer states at each worker, which prohibits training larger models with billions of parameters. To optimize memory usage, more advanced parallelism strategies must be employed. In this work, we propose general-purpose and theoretically principled adaptive batch size schedules compatible with data parallelism and model parallelism. We develop a practical implementation with PyTorch Fully Sharded Data Parallel, facilitating the pretraining of language models of different sizes. We empirically demonstrate that our proposed approaches outperform constant batch sizes and heuristic batch size warmup schedules in the pretraining of models in the Llama 2 family, with particular focus on smaller models with up to 3 billion parameters. We also establish theoretical convergence guarantees for such adaptive batch size schedules with ADAM for general smooth nonconvex objectives.

## 1. Introduction

Large-batch training (i.e., using large batch sizes) is arguably the current *de facto* training paradigm for large language models, driven by recent advances and the availability of computational hardware for deep learning. For instance, the open-weight model, Llama 3 405B [1], utilizes a batch size of 1024 sequences of length 4096, resulting in 4M tokens per batch. Despite the efficient utilization of available hardware through parallelization, a major drawback of large-batch training is the issue of "generalization gap" (see e.g., [2])—where model generalization performance deteriorates compared to small-batch training without heavy tuning of other hyperparameters. See Figure 1 for a graphical illustration of the existence of generalization gaps with different batch sizes when training a vanilla transformer with 61M parameters. Keskar et al. [3] argued that small-batch methods tend to converge to flat minima, leading to better generalization. To close this generalization gap, several works [4–6] have proposed using large learning rates to offset the effect of large batch sizes, recovering the

---

*Equal contribution

†Part of the work of Tim Tsz-Kit Lau and Mladen Kolar was performed when they were at The University of Chicago Booth School of Business.

Second Conference on Parsimony and Learning (CPAL 2025).

generalization performance of using small batches. However, the training of language (and vision) models based on the attention mechanism [7] and the transformer architecture is notoriously unstable. Reducing training instability, including unwanted loss spikes (see e.g., [8, 9]), demands significant tuning and cautious hyperparameter selections, like using a small learning rate.

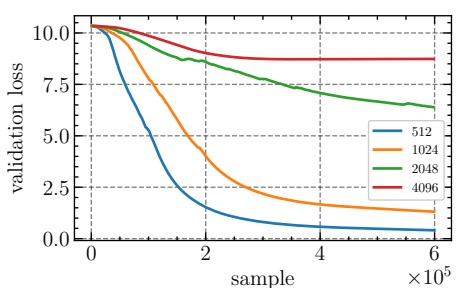

Figure 1: Generalization gap in transformer pretraining.

Beyond using a large learning rate to balance the intrinsic trade-off between training efficiency and generalization performance of large-batch training, Keskar et al. [3] also suggested the use of *adaptive sampling methods* [10, 11]. These methods are essentially adaptive batch size schemes that progressively improve the accuracy of the batch gradient approximation by gradually increasing batch sizes throughout the model training process. This concept has been explored by De et al. [12, 13] and Lau et al. [14], but their implementations are limited to the single-device setting, where all data samples are implicitly assumed to reside on the same device. This limitation makes them unfit for data-parallel distributed training wherein data is spread across various workers in a parallel system, potentially encompassing several network-connected nodes, thereby preventing the scaling necessary to train large models. Beyond the single-device setting, Lau et al. [15] have also extended such adaptive batch size schemes to local gradient methods for local batch sizes, where model synchronization is performed every several gradient steps rather than every step.

Data parallelism [16], such as `DistributedDataParallel` (DDP) in PyTorch [17] and counterparts in TensorFlow [18] and JAX [19, 20], is arguably the most popular paradigm for distributed training in deep learning. In data parallelism (alone), each worker holds a local copy of the model parameters (as well as gradient and optimizer states). The global input batch is divided into multiple minibatches for each training step, so each worker performs forward and backward computations with a different minibatch. After each training step, all GPUs perform an *all-reduce* collective communication to synchronize gradients, followed by a global model parameter update. This ensures that all local copies of the model remain identical after the parameter update steps. Adaptive batch size schemes can be developed based on the approaches in [10, 11, 21] for data-parallel settings, providing practical adaptive batch size schedules in PyTorch DDP for training large-scale deep neural networks, which require data parallelism.

While these practical schemes open up the possibility of distributed training of larger models with GPUs of lower memory, they are constrained by the inherent design of DDP—the need to maintain a model replica at each worker. State-of-the-art large language models (LLMs) now consist of billions or even hundreds of billions of parameters (e.g., Llama 3 405B [1]). Distributed training with only data parallelism thus unfortunately fails, as the memory required to store such models well exceeds the available memory of a single GPU. Even worse, access to expensive workstation-level GPUs with more memory is often limited to industrial labs, whereas academic researchers and end-users often have to resort to less powerful consumer-level GPUs or workstation-level GPUs with less memory.

To alleviate this limitation inherent to data parallelism, more memory-efficient paradigms of parallelism, such as model parallelism [22], have been proposed. In model parallelism, model parameters are sharded into various components and distributed to different workers. In particular, PyTorch Fully Sharded Data Parallel (FSDP) [23] is an implementation of model parallelism in PyTorch [24], marking the first native feature in PyTorch that can support models with up to trillions of parameters without relying on more sophisticated third-party libraries for model parallelism such as DeepSpeed [25], Megatron-LM [22, 26, 27], and their combinations [28], which could be overwhelming to get started with and too technical to modify for users' specific needs. Moreover, PyTorch FSDP has been widely adopted in the pretraining of various open-source language models such as OPT [29], TinyLlama [30], OLMo [31, 32], and DRBX [33].

However, even with data parallelism and model parallelism, LLM pretraining involving models with up to hundreds of billions of parameters and trillions of tokens (e.g., Llama 3 405B [1]), still incurs extensive costs (more than millions of US dollars per model) and imposes a significant carbon footprint. Consequently, there is a pressing need for developing proper and well-crafted training strategies. In this work, we focus on choosing dynamic batch size schedules, which deserve more attention than they have, since, unlike other optimizer hyperparameters, batch sizes also control training efficiency via memory utilization of GPUs, in addition to affecting model generalization performance and training stability. The current practice of choosing batch sizes in LLM pretraining, however, remains heuristic, in the sense that it usually involves either constant large batch sizes or prespecified heuristic warmup schedules which could be very hard to design.

**Contributions.** In this work, we propose theoretically principled adaptive batch size schedules based on the adaptive sampling method [10] for pretraining large language models, which are also generally applicable to training other deep neural networks. On the theoretical front, we establish a convergence guarantee for the proposed adaptive batch size schedules for ADAM, the *de facto* optimizer for pretraining language models. Various recent works have shown, both empirically and theoretically, that ADAM outperforms SGD in training attention-based language models [34–37]. Our convergence guarantee complements the existing results of adaptive batch size schedules for SGD [12, 13] and ADAGRAD [14]. From a practical perspective, we develop a solution of adaptive batch size schedules based on PyTorch FSDP, which are tailor-made for pretraining LLMs with more than billions of parameters.

## 2. Related Work

**Large-batch training of language models.** Large-batch training has proven to be very successful for different deep learning applications including computer vision [38, 39] and natural language processing [40–42]. From an empirical perspective, many open-source or open-weights models, such as OPT [29], BLOOM [43], Mistral 7B [44], Baichuan 2 [45], Qwen [46, 47], OLMo [31, 32], Gemma [48, 49], Llama [1, 50] and DeepSeek [51, 52], revealed that they were pretrained with large numbers of GPUs or TPUs (i.e., data-parallel sizes), hence naturally making use of large-batch training. While using large batch sizes is now standard, the rationale for choosing the magnitude of such large batch sizes is mostly based on hardware availability. Only recently in the training of Stable LM 2 1.6B, Bellagente et al. [53] clarified the selection of global batch sizes, aiming to strike an optimal balance between minimizing training time and the extra training tokens needed to reach the desired final training loss. Shallue et al. [54] study the effects of data parallelism by performing ablation studies on different batch sizes by training different models on different datasets using different optimizers, finding no evidence that large batch sizes degrade generalization performance with careful hyperparameter search. From a more theoretical perspective, McCandlish et al. [55] develop a model for understanding the *critical batch size* that determines the tradeoff between speed and efficiency of large-batch training. Kaplan et al. [56] further study the *scaling law* of the critical batch size as a power of the training loss only for language models. However, in most of these works, benchmarking was performed with different magnitudes of constant batch sizes, with the notable exception of McCandlish et al. [55] which provided a case study of dynamically varying the batch size with an adaptive batch size schedule, but only using a simple model (CNN) and dataset (SVHN). The effect of adaptive batch sizes for pretraining language models, to the best of our knowledge, remains elusive to the community.

**Batch size schedules.** Adaptive sampling methods [10, 11, 21], which adjust batch sizes based on gradient noise or gradient approximation quality, are further explored in deep learning [12–14, 57] but have not been applied to data parallelism with distributed samplers. The development of adaptive batch size schedules for deep learning is not a novel concept, featuring methodologies such as Big Batch SGD [12, 13], CABS [58], AdaBatch [59], SimiGrad [60] and AdaScale SGD [61]. Our work is also closely related to and motivated by the heuristic technique of batch size warmup/batch ramp, which has been widely adopted in pretraining LLMs and even in reinforcement learning [62]. Batch size warmup usually involves prespecified schedules of multiple batch size stages starting

from training with multiple increasing smaller batch sizes for small portions of the total training tokens, followed by training with the remaining tokens using a large batch size. For instance, GPT-3 [63] was pretrained by gradually increasing the batch size linearly from a small value (32k tokens) to the full value (3.2M tokens) over the first 4–12 billion tokens of training. Nemotron-4 [64] was pretrained with a batch size schedule of batch sizes 384–768–1152 sequences for 2.5%–2.5%–95% of the total number of training tokens. Llama 3 405B [1] was trained using the following batch size schedule: an initial batch size of 4M tokens with a sequence length 4096 tokens for 252M tokens; a batch size of 8M tokens with a sequence length of 8192 tokens for 2.87T tokens; a batch size of 16M tokens for the remainder of a total of about 15T training tokens. Such a batch size recipe is found to be able to stabilize training—few loss spikes were observed and it did not require interventions to correct for model training divergence. Despite potentially improving training efficiency or data parallelism, batch size warmup schedules remain heuristic and their impact on training is difficult to grasp. Another related yet seemingly orthogonal technique is sequence length warmup [65, 66], which progressively grows the sequence length throughout the pretraining process. Note that the pretraining of Llama 3 405B employs both batch size warmup and sequence length warmup.

## 3. Adaptive Batch Size Schedules with 2D Parallelism

We present the adaptive batch size schedules for data and model parallelism (termed *2D parallelism*), facilitating the scaling of pretraining for models with billions of parameters.

**Notation.** We define $[\![n]\!] := \{1, \ldots, n\}$ for $n \in \mathbb{N}^* := \mathbb{N} \setminus \{0\}$. We denote the inner product in $\mathbb{R}^d$ by $\langle \cdot, \cdot \rangle$ and its induced $L_2$-norm by $\|\cdot\|$, and $\|\cdot\|_1$ stands for the $L_1$-norm. For a vector $x \in \mathbb{R}^d$, $[x]_j$ denotes its $j$th coordinate ($j \in [\![d]\!]$). For a function $f \colon \mathbb{R}^d \to \mathbb{R} \cup \{\pm\infty\}$, $\partial_j f$ denotes its partial derivative with respect to its $j$th coordinate for $j \in [\![d]\!]$. The ceiling function is denoted by $\lceil \cdot \rceil$. The disjoint union of sets $\mathcal{S}_1, \ldots, \mathcal{S}_J$ is denoted by $\bigsqcup_{j \in [\![J]\!]} \mathcal{S}_j$.

### 3.1. Vanilla Adaptive Batch Size Schedules

We consider the empirical risk minimization problem in which we want to minimize the loss function $\mathcal{L} \colon \mathbb{R}^d \to \mathbb{R} \cup \{\pm\infty\}$ in the form of a finite-sum objective:

$$\underset{w \in \mathbb{R}^d}{\text{minimize}} \ \mathcal{L}(w) := \frac{1}{n} \sum_{i=1}^{n} \ell(w; z_i), \tag{1}$$

where $\ell \colon \mathbb{R}^d \times \mathcal{Z} \to \mathbb{R} \cup \{\pm\infty\}$ is the individual loss function, and $\mathcal{D}_n := \{z_i\}_{i=1}^n$ is the set of $n$ training samples. If $\ell(\cdot; z)$ is continuously differentiable for any $z \in \mathcal{Z}$, then the gradient of the loss function and its batch counterpart (i.e., the batch gradient) are given by

$$\nabla \mathcal{L}(w) := \frac{1}{n} \sum_{i=1}^{n} \nabla \ell(w; z_i) \quad \text{and} \quad \nabla \mathcal{L}_{\mathcal{B}}(w) := \frac{1}{b} \sum_{i \in \mathcal{B}} \nabla \ell(w; z_i),$$

where the batch $\mathcal{B} \subseteq [\![n]\!]$ is a subset of indices of data points sampled uniformly without replacement, and $b := |\mathcal{B}|$ is the corresponding batch size. We write $\ell_i(w) := \ell(w; z_i)$ and $\nabla \ell_i(w) := \nabla \ell(w; z_i)$. The batch gradient $\nabla \mathcal{L}_{\mathcal{B}}$ is used to approximate the full gradient $\nabla \mathcal{L}$ as the number of samples $n$ is prohibitively large.

**Norm test.** Falling into the family of adaptive sampling methods, the *norm test* [10] is motivated by measuring the quality of the approximation of the full gradient $\nabla \mathcal{L}$ by the batch gradient $\nabla \mathcal{L}_{\mathcal{B}}$ through the lens of approximation of a descent direction for the loss $\mathcal{L}$. If $\ell(\cdot; z)$ is also convex, then $-\nabla \mathcal{L}_{\mathcal{B}}$ is a descent direction for $\mathcal{L}$ at $w \in \mathbb{R}^d$ if and only if $\langle \mathcal{L}_{\mathcal{B}}(w), \mathcal{L}(w) \rangle \geqslant 0$, that is, $\mathcal{L}_{\mathcal{B}}$ and $\mathcal{L}$ share the same direction at $w$. It can be shown that the above inner product condition is equivalent to the norm condition: $\|\nabla \mathcal{L}_{\mathcal{B}}(w) - \nabla \mathcal{L}(w)\| \leqslant \eta \|\nabla \mathcal{L}(w)\|$ for any $\eta \in [0, 1)$. This condition cannot be checked directly, since the number of samples $n$ is in billions for LLMs and the full gradient $\nabla \mathcal{L}$ is unavailable. Instead, we have to resort to a batch approximation:

$$\frac{\|\mathrm{Var}_{i \in \mathcal{B}}(\nabla \ell_i(w))\|_1}{b} \cdot \frac{n - b}{n - 1} \leqslant \eta^2 \|\nabla \mathcal{L}_{\mathcal{B}}(w)\|^2, \tag{2}$$

where
$$\mathrm{Var}_{i\in\mathcal{B}}(\nabla\ell_i(w)) \coloneqq \frac{1}{b-1}\sum_{i\in\mathcal{B}}(\nabla\ell_i(w) - \nabla\mathcal{L}_{\mathcal{B}}(w))^2.$$

The adjustment factor $(n-b)/(n-1)$ is approximated by 1 as we take $n \to \infty$. Consequently, to ensure that the batch gradient approximates the descent direction of the full objective $\mathcal{L}$ well, the (*approximate*) *norm test* checks the following condition at each iteration $k \in \mathbb{N}^*$:

$$\frac{\|\mathrm{Var}_{i\in\mathcal{B}_k}(\nabla\ell_i(w_k))\|_1}{b_k} = \frac{1}{b_k(b_k-1)}\sum_{i\in\mathcal{B}_k}\left[\|\nabla\ell_i(w_k) - \nabla\mathcal{L}_{\mathcal{B}_k}(w_k)\|^2\right] \leqslant \eta^2\|\nabla\mathcal{L}_{\mathcal{B}_k}(w_k)\|^2, \quad (3)$$

and increases the next batch size $b_{k+1}$ if the above inequality is not satisfied, using

$$b_{k+1} = \left\lceil\frac{\|\mathrm{Var}_{i\in\mathcal{B}_k}(\nabla\ell_i(w_k))\|_1}{\eta^2\|\nabla\mathcal{L}_{\mathcal{B}_k}(w_k)\|^2}\right\rceil.$$

The condition can be viewed as an approximation of the following *exact variance norm test* in the stochastic setting:

$$\mathbb{E}_k\left[\|\nabla\mathcal{L}_{\mathcal{B}_k}(w_k) - \nabla\mathcal{L}(w_k)\|^2\right] \leqslant \eta^2\|\nabla\mathcal{L}(w_k)\|^2, \quad (4)$$

i.e., the motivating norm condition holds in expectation. Here $\mathbb{E}_k \coloneqq \mathbb{E}[\cdot\,|\,\mathcal{F}_k]$ denotes the conditional expectation with respect to the $\sigma$-algebra up to the current batch at iteration $k$, i.e., $\mathcal{F}_k \coloneqq \sigma(\{w_0, \mathcal{B}_0, \mathcal{B}_1, \ldots, \mathcal{B}_{k-1}\})$. After the next batch size is determined, the training loop continues with an optimizer step. The test implicitly makes a heuristic assumption that the next batch of size $b_{k+1}$ will satisfy the approximate norm test at the current iterate $w_k$, but this is never checked to streamline the training loop.

## 3.2. Adaptive Batch Size Schedules with Data Parallelism

To allow training with large batch sizes with parallelized computations, a data-parallel extension of the *norm test*, which is referred to as DDP-Norm, can be developed and can be implemented based on PyTorch DDP. A special treatment of the norm test with data parallelism is necessary since data samples now reside in different workers, but we need to compute the mean and the variance of all the per-sample gradients in the norm test.

Specifically, at each iteration $k$, the global batch $\mathcal{B}_k$ is split across $J$ workers with minibatches $(\mathcal{B}_{k,j})_{j\in[\![J]\!]}$ of equal size $b_{k,J}$ such that the global batch is the disjoint union of all minibatches, i.e., $\mathcal{B}_k = \bigsqcup_{j\in[\![J]\!]}\mathcal{B}_{k,j}$. Notice that at each worker $j \in [\![J]\!]$, the minibatch gradient can be computed by $\nabla\mathcal{L}_{\mathcal{B}_{k,j}}(w_k) = 1/b_{k,J}\sum_{i\in\mathcal{B}_{k,j}}\nabla\ell_i(w_k)$. Since the minibatches have equal size and are disjoint, applying the law of total expectation, the global batch gradient is equal to $\nabla\mathcal{L}_{\mathcal{B}_k}(w_k) = 1/J\sum_{j=1}^{J}\nabla\mathcal{L}_{\mathcal{B}_{k,j}}(w_k)$. Note that the averages across workers are computed using *all-reduce* operations in PyTorch DDP. When minibatch sizes exceed the maximum memory of the workers, the technique of gradient accumulation is applied to simulate larger global batch sizes.

It is worth noting that efficiently implementing the approximate norm test (3) in deep learning libraries such as PyTorch [24] is highly nontrivial, since per-sample gradients $\nabla\ell_i(w_k)$ are unavailable in the backward step of a standard training loop, but only the batch gradient $\nabla\mathcal{L}_{\mathcal{B}_k}(w_k)$ under a single-device setting or the minibatch gradient $\nabla\mathcal{L}_{\mathcal{B}_{k,j}}(w_k)$ at each worker $j$ under PyTorch DDP. If we were to implement the native approximate norm test (3), we would have had to compute per-sample gradients in parallel using vectorized mappings and based on a deep copy of the model, leading to undesirable memory and computational overheads. Thus, in practical implementation under data parallelism, instead of the approximate norm test (3), we propose to make use of the minibatch gradients of the workers to construct an estimator for the gradient variance

$$\widehat{\mathrm{Var}}_{i\in\mathcal{B}_k}(\nabla\ell_i(w_k)) \coloneqq \frac{1}{J}\sum_{j\in[\![J]\!]}\left(\nabla\mathcal{L}_{\mathcal{B}_{k,j}}(w_k) - \nabla\mathcal{L}_{\mathcal{B}_k}(w_k)\right)^2,$$

leading to the following more efficient implementation:

$$\frac{1}{b_k}\cdot\frac{1}{J}\sum_{j\in[\![J]\!]}\left[\|\nabla\mathcal{L}_{\mathcal{B}_{k,j}}(w_k) - \nabla\mathcal{L}_{\mathcal{B}_k}(w_k)\|^2\right] \leqslant \eta^2\|\nabla\mathcal{L}_{\mathcal{B}_k}(w_k)\|^2. \quad (5)$$

From now on, we refer the above alternative test as DDP-Norm. This implementation is much more computationally efficient since the minibatch gradients $\nabla \mathcal{L}_{\mathcal{B}_{k,j}}(w_k)$ are already available at each worker and the global batch gradient $\nabla \mathcal{L}_{\mathcal{B}_k}(w_k)$ can be computed using *all-reduce* operations. Note however that this implementation requires an additional *all-reduce* operation every time to compute the quantity on the left hand side of (5) and additional memory to store it.

### 3.3. Adaptive Batch Size Schedules with 2D Parallelism via PyTorch FSDP

To enable the training of models with more than billions of parameters, model-parallel training presents a more sophisticated paradigm of parallelism. It shards the parameters of models and allocates different shards to different workers. In essence, PyTorch FSDP [23], which shares similarities with ZeRO-3 [67, 68] in DeepSpeed [25], operates by substituting the *all-reduce* operation in PyTorch DDP with *all-gather* and *reduce-scatter* operations.

For the purpose of mathematical illustration, we focus particularly on the tensor parallelism aspect of model parallelism. Coupled with data parallelism, it is established that each worker $j$ possesses its own set of sharded parameters $\mathcal{W}_j$, $j \in [\![J]\!]$, such that all the model parameters are denoted by $w_k = (w_{k,j})_{j \in [\![J]\!]}$. Here, the sharded parameters on worker $j$ are represented by $w_{k,j} \in \mathcal{W}_j$. Consequently, to compute the microbatch gradient at worker $j$, the gradients of all parameter shards must be resharded to obtain $\nabla \mathcal{L}_{\mathcal{B}_{k,j}}(w_k) = (\nabla \mathcal{L}_{\mathcal{B}_{k,j}}(w_{k,1}), \ldots, \nabla \mathcal{L}_{\mathcal{B}_{k,j}}(w_{k,J}))$, which can be efficiently implemented using the API of PyTorch FSDP. The implementation of DDP-Norm based on PyTorch FSDP is referred to as FSDP-Norm.

## 4. Convergence Analysis

Complementary to the convergence results of the norm test for SGD [12, 13] and AdaGrad [14], we derive convergence guarantees for Adam, acknowledging its prevalence in training deep neural networks for both computer vision and, more recently, language models. Adam [69] employs the following update formula (with bias corrections for $m_k$ and $v_k$ dropped):

$$(\forall k \in \mathbb{N}^*) \quad m_k = \beta_1 m_{k-1} + (1-\beta_1)g_k, \quad v_k = \beta_2 v_{k-1} + (1-\beta_2)g_k^2, \quad w_{k+1} = w_k - \alpha m_k \odot v_k^{-1/2}, \quad (6)$$

where $g_k := \nabla \mathcal{L}_{\mathcal{B}_k}(w_k)$, $\alpha > 0$ is a constant learning rate, $(m_k)_{k \in \mathbb{N}^*}$ and $(v_k)_{k \in \mathbb{N}^*}$ are the sequences of exponential weighted moving averages of the first two moments of the batch gradients respectively, $(\beta_1, \beta_2) \in (0, \infty)^2$ are weighting parameters, $\odot$ denotes the Hadamard product, and the power operations are performed coordinate-wise. We omit the bias corrections of $m_k$ and $v_k$ to simplify the analysis, but note that it can be easily extended to incorporate bias corrections. We also consider the more challenging scenario where $v_k^{-1/2}$ instead of $(v_k^{1/2} + \varepsilon)^{-1}$ is used in the update, since the denominator of the adaptive step sizes is no longer lower bounded away from 0. In our analysis, we invoke the following assumptions.

**Assumption 1** ($L$-Lipschitz smoothness). The loss function $\mathcal{L}$ is $L$-Lipschitz smooth ($L > 0$): for any $(u, v) \in \mathbb{R}^d \times \mathbb{R}^d$, we have $\|\nabla \mathcal{L}(u) - \nabla \mathcal{L}(v)\| \leqslant L\|u - v\|$.

Similarly to the analysis for AdaGrad [14], we also require a coordinate-wise version of the (exact variance) norm test to hold due to the use of adaptive step size.

**Proposition 1.** The coordinate-wise (*exact variance*) *norm test with constant* $\eta \in (0, 1)$ *ensures that, for every iteration* $k \in [\![K]\!]$, *the coordinate-wise batch gradient* $\partial_i \mathcal{L}_{\mathcal{B}_k}(w_k)$ *satisfies the following* coordinate-wise expected strong growth (*E-SG*) *condition: for all* $i \in [\![d]\!]$, *we have*

$$\mathbb{E}_k[(\partial_i \mathcal{L}_{\mathcal{B}_k}(w_k))^2] \leqslant (1 + \eta^2)(\partial_i \mathcal{L}(w_k))^2.$$

Following closely a similar analysis to that in [70], we provide the following convergence results of the norm test for Adam.

**Theorem 1.** *Suppose that Assumption 1 holds. Let* $(w_k)_{k \in \mathbb{N}^*}$ *be the Adam iterates generated by* (6), *where the batch size* $b_k := |\mathcal{B}_k|$ *is chosen such that the coordinate-wise* (*exact variance*) *norm test with constant*

$\eta \in (0, 1)$ is satisfied at each iteration $k \in \mathbb{N}^*$. Then, if $0 < \beta_1 \leqslant \sqrt{\beta_2} - 8(1 + \eta^2)(1 - \beta_2)/\beta_2^2$ and $\beta_2 \in (0, 1)$, we have $\sum_{k=1}^{K} \mathbb{E}[\|\nabla \mathcal{L}(w_k)\|] \leqslant \widetilde{\mathcal{O}}(K)$, where $\widetilde{\mathcal{O}}$ hides any logarithmic factors.

The full statement of this theorem and its proofs, as well as more in-depth related discussions, are deferred to Appendix B. The convergence results presented do not account for the decoupled weight decay in AdamW [71], which is more commonly used as an optimizer for language model pretraining. Furthermore, considerations such as learning rate schedules and gradient clipping are not included in these findings. Extending the above convergence guarantees to these settings is highly challenging and nontrivial and is left for future work.

## 5. Numerical Experiments

To showcase the versatility and scalability of FSDP-Norm, we conduct experiments with various families of decoder-only autoregressive language models at with different sizes and pretraining datasets. These include MicroLlama 300M [72], TinyLlama 1.1B [30] and OpenLlama 3B [73] on the C4 dataset [74]. The C4 dataset are tokenized using the Llama 2 tokenizer [50] with a vocabulary size of 32,000. Experiments are conducted on workstations equipped with 4 NVIDIA L40S GPUs (MicroLlama) and 4 NVIDIA A100-SXM 80GB GPUs (TinyLlama and OpenLlama). The training of the latter two models only feasible with PyTorch FSDP but not with PyTorch DDP using such hardware configurations, even with mixed-precision training (bfloat16 is used). Our implementation utilizes the PyTorch FSDP API in PyTorch 2.6.1 and is simplified through Lightning Fabric of Lightning 2.4 [75]. For the ease of training language models, we also use LitGPT 0.5.3 [76]. Open-source implementation of DDP-Norm and FSDP-Norm is available at https://github.com/timlautk/adaptive-batch-fsdp.

**Training Specifications.** Adhering to the pretraining configurations of open-source LLMs such as TinyLlama [30] and OLMo [31], our training specifications include a linear warmup followed by a cosine decay learning rate schedule, and the AdamW optimizer with weight decay and gradient clipping. The adaptive batch size schedule is set to a maximum global batch size, above which the norm test is no longer performed, opting for fixed interval testing over step-by-step (a test interval 1 is used, but longer interval entails reduced overheads brought by the test). Efficiency dictates using the test in its original form rather than its coordinate-wise variant, despite convergence guarantees. Given that batch sizes increase to the maximum possible values in the early stages, we only pretrain our models for a number of samples that are sufficient to display the behavior of our method, treating these experiments mainly as proofs of concept. Detailed configurations are provided in Appendix C.

### 5.1. MicroLlama 300M

We first pretrain MicroLlama with 300M trainable parameters on the C4 dataset [74] under the same sets of other hyperparameters in order to better understand the effect of adaptive batch sizes. We compare with various constant batch size baselines $b_k \in \{2048, 4096, 8192\}$ and a stagewise batch size schedule 2048-4096-8192 for 2.5-2.5-95% of training tokens mimicking a popular batch size warmup for pretraining LLMs, and plot the results in Figure 2. We apply DDP-Norm for this relatively small model to demonstrate the applicability of the proposed schedules with PyTorch DDP. In Table 1, we report the total number of gradient steps (step), average batch size (bsz.), wall-clock time (time; in hours), best training loss (loss) and best validation loss (val loss; estimated by 100 iterations).

We observe from Figure 2 that with $\eta = 0.2$ or $\eta = 0.275$, our proposed DDP-Norm outperforms the constant batch size baselines by a large margin in terms of validation loss. Specifically, using the same number of training samples, from Table 1, our method achieves lower validation losses when using similar number of steps ($\eta = 0.2$ versus $b_k = 8192$), when we use the number of steps as the criterion of measuring training efficiency. Our proposed schedule with $\eta = 0.2$ performs slightly worse than the stagewise batch size schedule, but it is expected since the latter has a smaller averaged batch size and takes a larger number of training steps. It is also worth noting that the design of the stagewise schedule is completely heuristic and might require lots of tuning, e.g., the number of stages, values of batch sizes and their ratios.

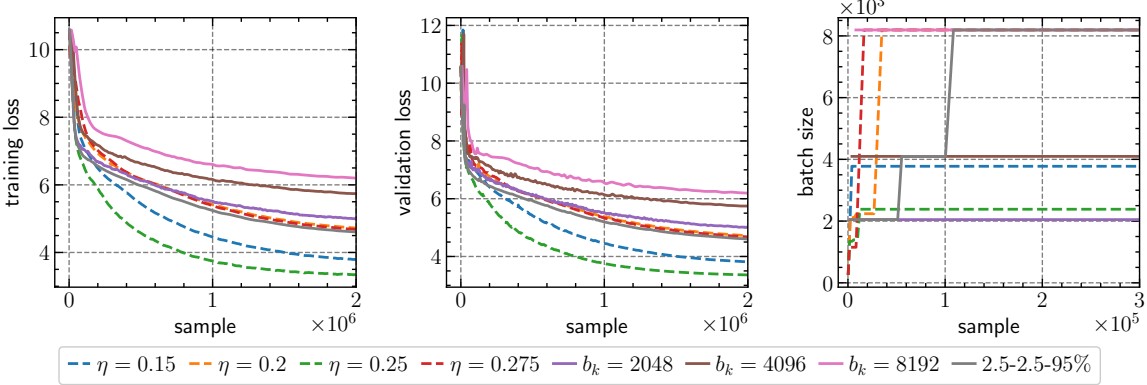

Figure 2: Training loss, validation loss and batch size schedule for MicroLlama 300M

| scheme | steps | bsz. | time | loss | val loss |
|---|---|---|---|---|---|
| $\eta = 0.15$ | 531 | 3770 | 9.31 | 3.764 | 3.811 |
| $\eta = 0.2$ | 254 | 7878 | 8.85 | 4.699 | 4.720 |
| $\eta = 0.25$ | 843 | 2373 | 10.30 | 3.313 | 3.361 |
| $\eta = 0.275$ | 252 | 7965 | 8.84 | 4.669 | 4.677 |
| $b_k = 2048$ | 977 | 2048 | 11.18 | 4.976 | 5.005 |
| $b_k = 4096$ | 489 | 4096 | 9.66 | 5.722 | 5.741 |
| $b_k = 8192$ | 245 | 8192 | 8.48 | 6.183 | 6.192 |
| 2.5-2.5-95% | 269 | 7439 | 8.78 | 4.594 | 4.604 |

Table 1: Results of MicroLlama 300M

We also observe that our method uses smaller batches at early stages and larger batches at later stages of training (e.g., $\eta \in \{0.2, 0.275\}$). This behavior has greater benefits regarding training efficiency because a larger batch size at each step means fewer number of required steps for the whole training process. On the other hand, our method greatly mitigates the side-effect of large-batch training—higher validation loss at the end of training—by starting from a small batch size and adaptively increasing it. Thus, our method enjoys both the good generalization performance of small batches and the high training efficiency of large batches. More importantly, our method is able to automatically increase batch sizes whenever it is necessary, to values that are completely adaptive to the training dynamics. Taking the adaptive batch size schedules in Figure 2 as an example, it is almost impossible to hand-craft similar schemes.

## 5.2. TinyLlama 1.1B

| scheme | steps | bsz. | time | loss | val loss |
|---|---|---|---|---|---|
| $\eta = 0.05$ | 261 | 7676 | 32.53 | 5.663 | 5.671 |
| $\eta = 0.075$ | 267 | 7521 | 32.67 | 5.705 | 5.704 |
| $\eta = 0.08$ | 270 | 7415 | 32.61 | 5.109 | 5.113 |
| $\eta = 0.085$ | 274 | 7312 | 32.83 | 4.257 | 4.256 |
| $b_k = 4096$ | 489 | 4096 | 34.48 | 3.814 | 3.817 |
| $b_k = 8192$ | 245 | 8192 | 32.41 | 4.895 | 4.893 |
| 2.5-2.5-95% | 269 | 7439 | 32.80 | 4.368 | 4.367 |

Table 2: Results of TinyLlama 1.1B

We also pretrain TinyLlama 1.1B on the C4 dataset, which necessitates the use of PyTorch FSDP and FSDP-Norm. From Figure 3 and Table 2, similar conclusions can be made. We observe that our proposed FSDP-Norm effectively narrows the generalization gap between large and small batches, compared with constant batch sizes and with stagewise batch size schedule baselines. Specifically, our method facilitates the adoption of larger batch sizes of 8192 during the later stages of training. For instance, our method with $\eta = 0.085$ achieves an averaged batch size of 7312, yet it achieves validation loss closer to that of $b_k = 4096$, compared to $b_k = 8192$. Our proposed method is also able to reduce the magnitude of potential loss spikes which are obvious in using constant batch sizes.

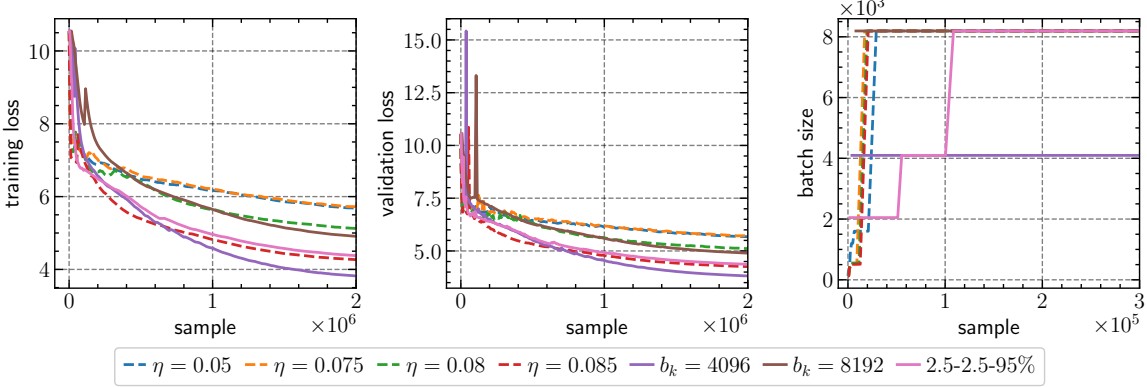

Figure 3: Training loss, validation loss and batch size schedule for TinyLlama 1.1B

## 5.3. OpenLlama 3B

| scheme | steps | bsz. | time | loss | val loss |
|---|---|---|---|---|---|
| $\eta = 0.05$ | 249 | 8045 | 19.54 | 4.943 | 4.935 |
| $\eta = 0.1$ | 253 | 7926 | 19.73 | 5.026 | 5.031 |
| $\eta = 0.15$ | 259 | 7726 | 19.59 | 4.549 | 4.554 |
| $b_k = 4096$ | 489 | 4096 | 20.75 | 3.934 | 3.956 |
| $b_k = 8192$ | 245 | 8192 | 19.53 | 5.113 | 5.104 |
| 2.5-2.5-95% | 269 | 7439 | 19.59 | 4.776 | 4.781 |

Table 3: Results of OpenLlama 3B

We finally pretrain OpenLlama 3B on the C4 dataset, where a shorter sequence length of 512 instead of 2048 is used due to constraint on compute resources. Again, we observe similar phenomena to those of the smaller models, as revealed in Figure 4 and Table 3. Specifically, with $\eta = 0.15$, the proposed approach requires slightly longer training time and larger number of training steps than the constant batch size 8192, while achieving a lower validation loss. While using a constant batch size 4096 achieves an even lower validation loss, it requires substantially more training steps and more than one hour of additional training time.

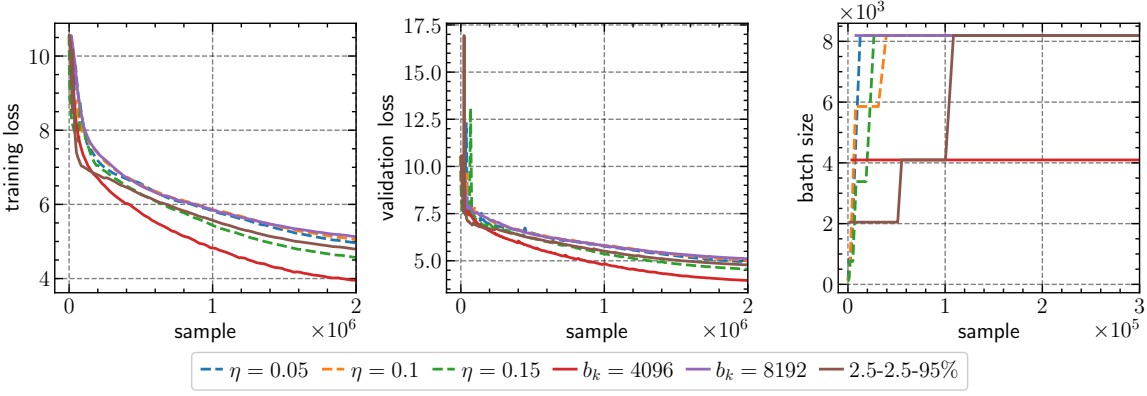

Figure 4: Training loss, validation loss and batch size schedule for OpenLlama 3B

## 5.4. Further Discussions of Experimental Results

**The effect of $\eta$.** The hyperparameter $\eta$ in the adaptive batch size schedules has the effect of controlling the probability of *obtaining a descent direction* and hence *increasing the batch size*. Obviously, choosing a right value of $\eta$ is vital for our method to succeed. Across all three sets of experiments of different model scales, we found that larger values of $\eta$ generally lead to more gradual batch size increments, but smaller values would allow full utilization of available compute resources at earlier stages of training but might defeat the prupose of adaptive batch sizes. Note that $\eta$ also varies with the

base learning rate $\alpha$ and the quality of the training datasets. In the series of works of adaptive sampling methods [10, 21, 77], there are in-depth discussions on choosing the learning rate via some line-search procedures, which are however usually infeasible when training large deep neural networks.

**Scaling law of critical batch size.** We conjecture that there are more general scaling laws of the critical batch size (see e.g., [56, 78–80]) in relation to $\eta$ which controls gradient approximation quality and the scale of gradient noise. For most choices of $\eta$ in the three sets of experiments, we choose $\eta$ small enough so that global batch sizes increase rapidly and reach the maximum possible values. However, in Figure 2, when $\eta = 0.15$, the final batch size is around 3800, which might be the critical batch size at this value of $\eta$. It is thus crucial to understand the notion of critical batch sizes through the lens of gradient approximation quality and we leave this for future work.

## 6. Concluding Remarks

We create an efficient PyTorch `FSDP` implementation of the norm test for large-scale distributed training, focusing on hardware use and ease of development. Our implementation shows that adaptive batch size schedules can pretrain Llama 2 language models with up to 3 billion parameters using few GPUs [50]. Furthermore, we provide convergence guarantees of the norm test for Adam, suggesting that our proposed adaptive batch size schedules are not only practically feasible, but also theoretically principled. Due to its generality, versatility, and scalability, we foresee extensive use of the adaptive batch size schedules in pretraining large transformer models like vision transformers (ViT) [81] and autoregressive image models (Aim) [82]. We emphasize our attention on a PyTorch `FSDP` approach due to its integration with PyTorch. However, a more advanced implementation of the adaptive batch size schedules, using a new version of PyTorch `FSDP` (`FSDP2`) and tensor parallelism via PyTorch `DTensor` (Distributed Tensor), as well as availing of stronger computational hardware, will significantly enhance the scalability of the method for training models exceeding 7B parameters with 2D, 3D or even 4D parallelism. For further exploration, we refer readers to the `torchtitan` [83] and `lingua` [84] repositories. Furthermore, while our current implementation is based on PyTorch `FSDP`, but is readily extendable to other deep learning frameworks such as JAX [19] with `FSDP` and/or GShard [85].

**Limitations.** In this work, we are primarily concerned with model generalization performance measured by validation loss without any evaluation on downstream benchmarks. The main reason for this is that we did not fully pretrain the models for sufficient number of tokens, implying that these models will not be competitive on downstream benchmarks. However, we expect that models fully pretrained with our proposed schedules will achieve very competitive performance on the evaluation of downstream benchmarks. We also remark that we can also incorporate other paradigms of parallelism such as pipeline and context parallelism with our proposed scheme, leading to 4D parallelism (data, tensor, pipeline, context parallel) for large-scale pretraining. While not supported in our current implementation, this can be achieved using the recent library `picotron` [86] or the more sophisticated library Megatron-LM [22]. We leave this implementation for future work.

**Acknowledgments**

Tim Tsz-Kit Lau would like to thank Zhihan Zhou for helpful discussion on the experimental setup. The research of Han Liu is supported by NIH R01LM01372201, NSF RI 1840857, NSF TRIPOD 1740735, NSF DMS1454377-CAREER, NSF IIS 1546482, along with an Alfred P. Sloan Fellowship. The research of Mladen Kolar is supported in part by NSF ECCS-2216912. This research is supported in part through the computational resources and staff contributions provided for the Quest high performance computing facility at Northwestern University which is jointly supported by the Office of the Provost, the Office for Research, and Northwestern University Information Technology. This research is also supported in part through the computational resources and staff contributions provided for the Data Science Institute at the University of Chicago, through its AI + Science Research Funding Initiatives.

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

# APPENDIX

# Contents

# A. Additional Details of The Proposed Algorithm

Note that the use of PyTorch FSDP does not lead to significance difference in the implementation of the norm test compared to its DDP implementation. We assume that the gradients of different parameter shards are concatenated together in the following computation to simplify the representation.

## A.1. The Overall Algorithm

---

**Algorithm 1** DDP-NORM or FSDP-NORM for ADAMW

---

**Input:** $w_1 \in \mathbb{R}^d$, $m_0 = v_0 = \mathbf{0}_d \in \mathbb{R}^d$, $(\alpha, \lambda, \varepsilon, \beta_1, \beta_2) \in (0, \infty)^5$, $\mathcal{D}_n = \{z_i\}_{i \in [\![n]\!]} \subset \mathcal{Z}$, number of workers $J \in \mathbb{N}^*$, number of gradient accumulation steps $M \in \mathbb{N}^*$, number of training samples $N \in \mathbb{N}^*$, step counter $k = 1$, processed sample counter $i = 0$, initial (global) batch size $b_0$, initial microbatch size $b_{0,J}^M = b_0/(JM)$

  **while** $i < N$ **do**

    Sample the i.i.d. data batch (indices) $\mathcal{B}_k$ uniformly from $[\![n]\!]$ of size $b_k := |\mathcal{B}_k|$

    Split $\mathcal{B}_k$ evenly to each worker $j \in [\![J]\!]$, each with $\mathcal{B}_{k,j}$ of size $b_{k,J}$

    **for all** $j = 1, \ldots, J$ in parallel **do**

      Split $\mathcal{B}_{k,j}$ evenly to each gradient accumulation step $m \in [\![M]\!]$, each with $\mathcal{B}_{k,j}^m$ of size $b_{k,J}^M$

      Initialize $\nabla \mathcal{L}_{\mathcal{B}_{k,j}}(w_k) = \mathbf{0}_d$

      **for** $m = 1, \ldots, M$ **do**

        Compute $\frac{1}{M} \nabla \mathcal{L}_{\mathcal{B}_{k,j}^m}(w_k)$

        Accumulate gradients $\nabla \mathcal{L}_{\mathcal{B}_{k,j}}(w_k) \leftarrow \nabla \mathcal{L}_{\mathcal{B}_{k,j}}(w_k) + \frac{1}{M} \nabla \mathcal{L}_{\mathcal{B}_{k,j}^m}(w_k)$

      **end for**

    **end for**

    Compute the batch gradient $g_k := \nabla \mathcal{L}_{\mathcal{B}_k}(w_k)$ with *all-reduce*

    Compute the approximate gradient variance $\widehat{\mathrm{Var}}_{i \in \mathcal{B}_k}(\nabla \ell_i(w_k))$ with *all-reduce*

$$\widehat{\mathrm{Var}}_{i \in \mathcal{B}_k}(\nabla \ell_i(w_k)) := \frac{1}{J} \sum_{j \in [\![J]\!]} \left( \nabla \mathcal{L}_{\mathcal{B}_{k,j}}(w_k) - g_k \right)^2$$

    Compute the approximate norm test statistic

$$\mathsf{T}_k \equiv \mathsf{T}(w_k; \mathcal{B}_k, \eta) := \frac{\left\| \widehat{\mathrm{Var}}_{i \in \mathcal{B}_k}(\nabla \ell_i(w_k)) \right\|_1}{\eta^2 \|g_k\|^2}$$

    **if** $\mathsf{T}_k > b_k$ **then**

      Increase the next global batch size $b_{k+1} = \lceil \mathsf{T}_k \rceil$

      Round up the microbatch size $b_{k+1,J}^M = \lceil b_{k+1}/(JM) \rceil$

      Update the minibatch size $b_{k+1,J} = M b_{k+1,J}^M$

      Update the global batch size again $b_{k+1} = J b_{k+1,J}$

    **else**

      $b_{k+1} = b_k$

    **end if**

    $m_k = \beta_1 m_{k-1} + (1 - \beta_1) g_k$            ▷ *ADAMW*

    $v_k = \beta_2 v_{k-1} + (1 - \beta_2) g_k^2$

    $\widehat{m}_k = m_k \odot (1 - \beta_1^k)^{-1}$

    $\widehat{v}_k = v_k \odot (1 - \beta_2^k)^{-1}$

    $w_{k+1} = (1 - \alpha\lambda) w_k - \alpha \widehat{m}_k \odot (\widehat{v}_k^{1/2} + \varepsilon)^{-1}$

    $k \leftarrow k + 1$

    $i \leftarrow i + b_k$

  **end while**

---

# B. Proofs of Main Text

We give a brief sketch of the omitted proof of the main text in this section. Notice that the convergence analysis of the norm test for ADAM largely follows that in [70], where more details and remarks of the analysis and rationales of its derivation can be found.

*Remark* B.1. Despite the similarity of the proof techniques, we emphasize that our setting requires less restrictive assumptions. While Wang et al. [70] assume the stochastic oracle of the gradient (i.e., batch gradient in our case) has coordinate-wise affine variance, i.e., for any batch of samples $\mathcal{B} \subseteq \mathcal{D}_n$ and $(\sigma, \tau) \in (0, \infty)^2$, we have

$$(\forall i \in [\![d]\!])(\forall w \in \mathbb{R}^d) \quad \mathbb{E}\big[(\partial_i \mathscr{L}_{\mathcal{B}}(w))^2\big] \leqslant \sigma^2 + \tau^2 (\partial_i \mathscr{L}(w))^2.$$

We do not impose this global condition which is often difficult to verify in practical scenarios, but instead we increase the (next) batch size such that the condition of the coordinate-wise (exact variance) norm test with constant $\eta \in (0, 1)$ is satisfied at the current iterate $w_k \in \mathbb{R}^d$ with the current batch $\mathcal{B}_k \subseteq \mathcal{D}_n$:

$$(\forall i \in [\![d]\!]) \quad \mathbb{E}_k\big[(\partial_i \mathscr{L}_{\mathcal{B}_k}(w_k) - \partial_i \mathscr{L}(w_k))^2\big] \leqslant \eta^2 (\partial_i \mathscr{L}(w_k))^2,$$

which implies

$$(\forall i \in [\![d]\!]) \quad \mathbb{E}_k\big[(\partial_i \mathscr{L}_{\mathcal{B}_k}(w_k))^2\big] \leqslant (1 + \eta^2)(\partial_i \mathscr{L}(w_k))^2,$$

which is also known as the coordinate-wise *expected strong growth* (E-SG) condition [14]. Note that the coordinate-wise (E-SG) condition implies the coordinate-wise *relaxed growth* (RG) condition [87], adopting the nomenclature in [88]:

$$(\forall i \in [\![d]\!]) \quad \mathbb{E}_k\big[(\partial_i \mathscr{L}_{\mathcal{B}_k}(w_k))^2\big] \leqslant \sigma^2 + \tau^2 (\partial_i \mathscr{L}(w_k))^2,$$

where $\tau^2 = 1 + \eta^2$ and $\sigma \in (0, \infty)$. Recall that we only require such a condition to hold at the current iterate $w_k$ with the current batch $\mathcal{B}_k$, through the enforcement of the coordinate-wise (exact variance) norm test. Even though the exact variance test is not implemented in practice but its approximate version instead, this is often a good heuristic to justify the convergence of the test.

**Additional notation.** To simplify notation, we denote the full gradient by $\mathscr{G}_k := \nabla \mathscr{L}(w_k)$, and its $i$th coordinate by $\mathscr{G}_{k,i} := \partial_i \mathscr{L}(w_k)$.

## B.1. Technical Lemmas

We state without proof the following technical lemmas from [70].

**Lemma B.1.** *Let $0 < \beta_1^2 < \beta_2 < 1$ and consider a sequence of real numbers $(a_n)_{n \in \mathbb{N}^*} \subset \mathbb{R}$. Let $b_0 > 0$, $b_k = \beta_2 b_{k-1} + (1 - \beta_2) a_k^2$, $c_0 = 0$ and $c_k = \beta_1 c_{k-1} + (1 - \beta_1) a_k$. We have the following inequality*

$$\sum_{k=1}^K \frac{|c_k|^2}{b_k} \leqslant \frac{(1-\beta_1)^2}{(1-\beta_2)(1-\beta_1/\sqrt{\beta_2})^2}\left(\log\left(\frac{b_K}{b_0}\right) - K \log \beta_2\right). \tag{B.1}$$

**Lemma B.2.** *Consider the ADAM iterates $(w_k)_{k \in \mathbb{N}^*}$ generated by (6). Then we have*

$$(\forall k \in \mathbb{N}^*) \quad |w_{k+1,i} - w_{k,i}| \leqslant \alpha \frac{1-\beta_1}{\sqrt{1-\beta_2}\sqrt{1-\beta_1^2/\beta_2}} \leqslant \alpha \frac{1-\beta_1}{\sqrt{1-\beta_2}\sqrt{1-\beta_1/\beta_2}}.$$

*Proof Sketch.* This is due to the definition of the ADAM iterate and Cauchy–Schwarz's inequality. □

## B.2. Proof of Theorem 1

Since the proof of Theorem 1 is highly similar to that in [70], we just provide a proof sketch. We state the formal theorem as follows.

**Theorem B.1** (Formal version of Theorem 1). *Suppose that Assumption 1 holds. Let $(w_k)_{k\in\mathbb{N}^*}$ be the ADAM iterates generated by (6), where the batch size $b_k := |\mathcal{B}_k|$ is chosen such that the coordinate-wise (exact variance) norm test with constant $\eta \in (0,1)$ is satisfied at each iteration $k \in \mathbb{N}^*$. Then, if $0 < \beta_1 \leqslant \sqrt{\beta_2} - 8(1+\eta^2)(1-\beta_2)/\beta_2^2$ and $\beta_2 \in (0,1)$, we have*

$$\sum_{k=1}^{K} \mathbb{E}[\|\nabla\mathcal{L}(w_k)\|]$$

$$\leqslant \sqrt{c_2 + 2c_1 \sum_{i=1}^{d}\left[\log\left(2(K+1)\sum_{i=1}^{d}\sqrt{v_{0,i}+\sigma^2} + 24d\frac{\tau^2 c_1}{\sqrt{\beta_2}}\log\left(d\frac{\tau^2 c_1}{\sqrt{\beta_2}}\right) + \frac{12\tau^2}{\sqrt{\beta_2}}c_2\right)\right]}$$

$$\times \sqrt{2(K+1)\sum_{i=1}^{d}\sqrt{v_{0,i}+\sigma^2} + 24d\frac{\tau^2 c_1}{\sqrt{\beta_2}}\log\left(d\frac{\tau^2 c_1}{\sqrt{\beta_2}}\right) + \frac{12\tau^2}{\sqrt{\beta_2}}c_2}, \quad \text{(B.2)}$$

*where $v_{0,i}$ is the $i$-coordinate of $v_0$, $\tau^2 = 1+\eta^2$, $\sigma \in (0,\infty)$,*

$$c_1 := \frac{32L\alpha\left(1+\beta_1/\sqrt{\beta_2}\right)^3}{(1-\beta_2)\left(1-\beta_1/\sqrt{\beta_2}\right)^3} + \frac{16\beta_1^2\sigma(1-\beta_1)}{\beta_2\sqrt{1-\beta_2}\left(1-\beta_1/\sqrt{\beta_2}\right)^3} + \frac{64(1+\sigma^2)\sigma^2 L^2\alpha^2 d}{\beta_2^2\left(1-\beta_1/\sqrt{\beta_2}\right)^4\sigma(1-\beta_2)^{3/2}},$$

$$c_2 := \frac{8(1-\beta_1/\sqrt{\beta_2})}{\alpha(1-\beta_1)} + \frac{32}{\beta_2\left(1-\beta_1/\sqrt{\beta_2}\right)^2}\sum_{i=1}^{d}\mathbb{E}\left[\frac{\mathcal{G}_{1,i}^2}{\sqrt{\tilde{v}_{1,i}}}\right] + 2c_1\sum_{i=1}^{d}\left(\log\left(\frac{1}{\sqrt{\beta_2 v_{0,i}}}\right) - K\log\beta_2\right),$$

$$u_k := \frac{w_k - \beta_1 w_{k-1}/\sqrt{\beta_2}}{1-\beta_1/\sqrt{\beta_2}}.$$

The proof consists of deriving a descent lemma on the sequence $u_k := \frac{w_k - \beta_1 w_{k-1}/\sqrt{\beta_2}}{1-\beta_1/\sqrt{\beta_2}}$.

**Lemma B.3.** *Suppose that all the assumptions in Theorem B.1 hold. We also define the function $\varphi_k := \mathbb{E}\left[-\alpha\left\langle\mathcal{G}_k, \mathcal{G}_k \odot \tilde{v}_{k+1}^{-1/2}\right\rangle\right]$. Then we have*

$$\mathbb{E}[\mathcal{L}(u_{k+1})]$$

$$\leqslant \mathbb{E}[\mathcal{L}(u_k)] - \frac{\alpha(1-\beta_1)}{4(1-\beta_1/\sqrt{\beta_2})}\mathbb{E}\left[-\alpha\left\langle\mathcal{G}_k, \mathcal{G}_k \odot \tilde{v}_k^{-1/2}\right\rangle\right] + \frac{2\alpha\sigma\sqrt{1-\beta_2}}{(1-\beta_1^2/\beta_2)^2}\sum_{i=1}^{d}\left[\frac{g_{k,i}^2}{v_{k,i}}\right]$$

$$+ \frac{4\alpha\tau^2}{(1-\beta_1/\sqrt{\beta_2})^2\sqrt{\beta_2}}\sum_{i=1}^{d}\mathbb{E}\left[\frac{1}{\beta_2}\varphi_{k-1} - \varphi_k\right] + \sum_{i=1}^{d}\frac{2\alpha\sigma\sqrt{1-\beta_2}}{(1-\beta_1)(1-\beta_1/\sqrt{\beta_2})}\mathbb{E}\left[\frac{m_{k,i}^2}{v_{k,i}}\right]$$

$$+ \frac{64d(1+\tau^2)\tau^2 L^2\alpha^3}{\beta_2^2(1-\beta_1/\sqrt{\beta_2})^3(1-\beta_1)\sigma\sqrt{1-\beta_2}} \cdot \mathbb{E}\left[\left\|m_{k-1}\odot v_{k-1}^{-1/2}\right\|^2\right]$$

$$+ \sum_{i=1}^{d}\frac{2\alpha\sigma\beta_1^2\sqrt{1-\beta_2}}{\beta_2(1-\beta_1)(1-\beta_1/\sqrt{\beta_2})}\mathbb{E}\left[\frac{m_{k-1,i}^2}{v_{k-1,i}}\right]$$

$$+ L\mathbb{E}\left[4\alpha^2\left(\frac{\beta_1/\sqrt{\beta_2}}{1-\beta_1/\sqrt{\beta_2}}\right)^2\left\|m_{k-1}\odot v_{k-1}^{-1/2}\right\|^2 + 3\alpha^2\left(\frac{1}{1-\beta_1/\sqrt{\beta_2}}\right)^2\left\|m_k\odot v_k^{-1/2}\right\|^2\right].$$

*Proof Sketch.* This bound is derived by bounding the "first-order term" and the "second-order term", similar to the derivation of a descent lemma for Lipschitz smooth functions but on the sequence $(u_k)_{k\in\mathbb{N}^*}$. □

**Lemma B.4.** *Suppose that all the assumptions in Theorem B.1 hold. Then we have*

$$\sum_{k=1}^{K+1}\sum_{i=1}^{d}\mathbb{E}[\tilde{v}_{k,i}^{1/2}] \leqslant 2(K+1)\sum_{i=1}^{d}\sqrt{v_{0,i}+\sigma^2} + \frac{24d\tau^2 c_1}{\sqrt{\beta_2}}\log\left(\frac{d\tau^2 c_1}{\sqrt{\beta_2}}\right) + \frac{12\tau^2 c_2}{\sqrt{\beta_2}}.$$

*Proof Sketch.* This bound is derived by a *divide-and-conquer* approach, considering the cases $|\mathcal{G}_{k,i}| \geqslant \sigma/\tau$ and $|\mathcal{G}_{k,i}| \leqslant \sigma/\tau$ respectively. □

*Proof Sketch of Theorem 1.* The final bound is derived by first summing the inequality in Lemma B.3 with the assumed condition of $(\beta_1, \beta_2)$. Further application of Lemma B.1, Cauchy–Schwarz's inequality and Lemma B.4 implies the desired result. □

## C. Details of Numerical Experiments

We provide a summary table for the architecture of the language models we pretrained. More details of these models can be found in [30, 72, 73].

| Model | MicroLlama 300M | TinyLlama 1.1B | OpenLlama 3B |
|---|---|---|---|
| $n_{\text{params}}$ | 304.6M | 1.1B | 3.4B |
| $d_{\text{model}}$ | 2048 | 2048 | 2048 |
| $n_{\text{layers}}$ | 12 | 22 | 26 |
| $n_{\text{heads}}$ | 12 | 32 | 32 |
| $d_{\text{head}}$ | 64 | 64 | 100 |

Table 4: Specifications of models

We also summarize the training hyperparameters of the three sets of experiments in the following tables.

### C.1. MicroLlama 300M

| Model | MicroLlama 300M |
|---|---|
| Training samples (sequences) | 2000000 |
| Learning rate schedule | Linear warmup + cosine decay |
| Learning rate warmup (samples) | 20000 (1% of training samples) |
| Sequence length (tokens) | 2048 |
| Optimizer | ADAMW |
| Optimizer scaling rule | None |
| $(\beta_1, \beta_2)$ | $(0.9, 0.95)$ |
| $\varepsilon$ | $10^{-8}$ |
| Peak learning rate | 0.0004 |
| Minimum learning rate | 0.00004 |
| Base micro batch size | 4 |
| Maximum micro batch size | 8 |
| Base global batch size | 256 |
| Maximum global batch size | 8192 |
| Base gradient accumulation steps | 16 |
| Data-parallel size | 4 |
| Weight decay | 0.1 |
| Weight decay skip bias | No |
| Precision | `bfloat16` |
| Gradient clipping | 1.0 |
| Test interval | 1 |

Table 5: Training hyperparameters for MicroLlama 300M

## C.2. TinyLlama 1.1B

| Model | TinyLlama 1.1B |
| --- | --- |
| Training samples (sequences) | 2000000 |
| Learning rate schedule | Linear warmup + cosine decay |
| Learning rate warmup (samples) | 20000 (1% of training samples) |
| Sequence length (tokens) | 2048 |
| Optimizer | AdamW |
| Optimizer scaling rule | None |
| $(\beta_1, \beta_2)$ | $(0.9, 0.95)$ |
| $\varepsilon$ | $10^{-8}$ |
| Peak learning rate | 0.0004 |
| Minimum learning rate | 0.00004 |
| Base micro batch size | 4 |
| Maximum micro batch size | 8 |
| Base global batch size | 128 |
| Maximum global batch size | 8192 |
| Base gradient accumulation steps | 16 |
| Data-parallel size | 4 |
| Weight decay | 0.1 |
| Weight decay skip bias | No |
| Precision | `bfloat16` |
| Gradient clipping | 1.0 |
| Test interval | 1 |

Table 6: Training hyperparameters for TinyLlama 1.1B

## C.3. OpenLlama 3B

| Model | OpenLlama 3B |
|---|---|
| Training samples (sequences) | 2000000 |
| Learning rate schedule | Linear warmup + cosine decay |
| Learning rate warmup (samples) | 20000 (1% of training samples) |
| Sequence length (tokens) | 512 |
| Optimizer | ADAMW |
| Optimizer scaling rule | None |
| $(\beta_1, \beta_2)$ | $(0.9, 0.95)$ |
| $\varepsilon$ | $10^{-8}$ |
| Peak learning rate | 0.0004 |
| Minimum learning rate | 0.00004 |
| Base micro batch size | 4 |
| Maximum micro batch size | 8 |
| Base global batch size | 128 |
| Maximum global batch size | 8192 |
| Base gradient accumulation steps | 16 |
| Data-parallel size | 4 |
| Weight decay | 0.1 |
| Weight decay skip bias | No |
| Precision | bfloat16 |
| Gradient clipping | 1.0 |
| Test interval | 1 |

Table 7: Training hyperparameters for OpenLlama 3B

