# OpenReview forum: "Adaptive Batch Size Schedules for Distributed Training of Language Models with Data and Model Parallelism"
_CPAL.cc/2025/Proceedings_Track — CPAL 2025 (Proceedings Track) Poster_

### Official Review · Reviewer_XwPF · 2025-01-12
**Review of Submission 62**

**Rating:** 5
**Confidence:** 3

**Review:**

This work propose a adaptive batch size strategy that compatible with 2D-parallel distributed training.

(+) The proposed method is evaluated both empirically and theoretically.

(+) Plenty of experiments are conducted for demonstrating the effectivenss of the proposed method.

(+) The manuscript is logically organized and easy to follow.

(-) The propose method seem like a extenson of norm-test method to 2-D distribution training, making the novelty modest, it's better to discuss the advantages against the baseline methods discussed in Line 71 and demonstrate more about the unique challenges of applying adaptive batch-size method to distribution training.

(-) Does the method also compatible with 3-D distribution training (data/model/tensor-parallellism)?

(-) As shown in Figure 1, larger batch size results worse generalization, is it because the whole training set size is relatively small? thus the optimization step becomes smaller for larger batch-size training, if the training set is sufficient large, whether larger batch size still achieve worse performance?

---

### Official Review · Reviewer_Mo8A · 2025-01-13
**Review for submission 62**

**Rating:** 6
**Confidence:** 3

**Review:**

**Summary:**

This paper proposes adaptive batch size schedules for distributed training of large-scale language models using both data and model parallelism. It introduces a method compatible with FSDP, tensor, and data parallelism, along with theoretical analysis demonstrating scalability and improved efficiency in pretraining language models with up to 3 billion parameters. Experimental evaluations demonstrate that the approach outperforms traditional constant or heuristic batch size schedules, offering convergence guarantees and practical applicability for large-scale deep learning tasks.

**Pros:**
- The paper is well-written and well-motivated overall.
- The proposed adaptive batch size training algorithm is intuitive and easy to understand.
- Both theoretical and experimental results are provided to justify the performance of the method.

**Cons:**
- Only the loss is shown to justify the performance of models in the experiments. However, loss alone does not fully reflect model quality. It would be helpful to include evaluation benchmark results, such as MMLU or GSM8k.
- The proposed method is motivated by its compatibility with both data and tensor parallelism. However, there is no clear reason why the method needs to be specifically adapted for tensor model parallelism.
- It is unclear if the method can be easily adapted to modern parallelism techniques like pipeline and context parallelism (sometimes referred to as ring attention). A discussion on this topic would add value.

**Minor Questions & Comments:**
- Including an algorithm box for the proposed method would improve clarity.
- Is there really a model called Olmo-65B? Upon checking [1], only 1B and 3B models appear to exist.

---

### Official Review · Reviewer_bQkT · 2025-01-14

**Rating:** 7
**Confidence:** 3

**Review:**

Strengths

- The paper proposes theoretically-principled adaptive batch size schedules based on the adaptive sampling norm test for pretraining large language models.
- The authors develop efficient implementations of their adaptive batch size schedules using PyTorch Fully Sharded Data Parallel (FSDP), enabling scaling to models with billions of parameters by leveraging both data and model parallelism.
- Good storyline with solid results

Suggestions

- The convergence analysis does not account for important practical considerations like decoupled weight decay in AdamW, learning rate schedules, and gradient clipping. Extending the theoretical guarantees to cover these aspects, while challenging, would make the results more comprehensive and reflective of real-world training settings.

- The related work section discusses several existing adaptive batch size methods for deep learning, but the experiments do not include comparisons to these baselines. Empirical comparisons would help better position the proposed approach against prior work.

- The experiments focus on decoder-only autoregressive language models. Discussing the applicability of the approach to other architectures like encoder-decoder models, or even other domains like vision, would be helpful to emphasize the generality of the adaptive batch size schedules.

---

### Meta-Review · Area_Chair_fSu1 · 2025-02-04

**Recommendation:** Accept (Poster)
**Confidence:** 4

**Metareview:**

The paper introduces a theoretically motivated adaptive batch size scheduling method for distributed training of large-scale language models, demonstrating compatibility with data and model parallelism through FSDP-based implementation. The approach is supported by both theoretical analysis and extensive empirical validation, showcasing improved scalability and efficiency for models with billions of parameters. Data and model parallelism development is especially fundamental to applying training algorithms. I believe that this work should be accepted.

---

### Decision · Program_Chairs · 2025-02-11

Accept (Poster)